# Red-Edge Excitation Shift Spectroscopy (REES): Application to Hidden Bound States of Ligands in Protein–Ligand Complexes

**DOI:** 10.3390/ijms22052582

**Published:** 2021-03-04

**Authors:** Md Lutful Kabir, Feng Wang, Andrew H. A. Clayton

**Affiliations:** 1Optical Sciences Center, Department of Physics and Astronomy, School of Science, Swinburne University of Technology, Melbourne, VIC 3122, Australia; mdlutfulkabir@swin.edu.au; 2Center for Translational Atomaterials, Department of Chemistry and Biotechnology, School of Science, Swinburne University of Technology, Melbourne, VIC 3122, Australia; fwang@swin.edu.au

**Keywords:** fluorescence spectroscopy, red-edge excitation shift, protein–ligand interactions, free energy landscape, kinase inhibitors

## Abstract

Ligand-protein binding is responsible for the vast majority of bio-molecular functions. Most experimental techniques examine the most populated ligand-bound state. The determination of less populated, intermediate, and transient bound states is experimentally challenging. However, hidden bound states are also important because these can strongly influence ligand binding and unbinding processes. Here, we explored the use of a classical optical spectroscopic technique, red-edge excitation shift spectroscopy (REES) to determine the number, population, and energetics associated with ligand-bound states in protein–ligand complexes. We describe a statistical mechanical model of a two-level fluorescent ligand located amongst a finite number of discrete protein microstates. We relate the progressive emission red shift with red-edge excitation to thermodynamic parameters underlying the protein–ligand free energy landscape and to photo-physical parameters relating to the fluorescent ligand. We applied the theoretical model to published red-edge excitation shift data from small molecule inhibitor–kinase complexes. The derived thermodynamic parameters allowed dissection of the energetic contribution of intermediate bound states to inhibitor–kinase interactions.

## 1. Introduction

Ligand–protein binding is a key regulatory process that controls the vast majority of bio-molecular functions [1]. Evolution has shaped the constituents of living matter so that these macromolecular recognition events are largely robust and specific. On the other hand, disease states can result from improper protein–ligand interactions, which drive malfunctions in cellular behavior. Ligand–protein binding is also of interest to drug companies, where often the goal is to develop drugs that bind with increasing affinity and specificity to a particular drug target. These companies also use structural techniques such as x-ray crystallography or Nuclear magnetic resonance spectroscopy (NMR) to characterize the structure of the ligand–protein complex and to define the environment surrounding the ligand in the protein-binding pocket. Structure-based design aims to increase the affinity of the drug for the drug target based on this structural knowledge [2].

Drug efficacy, on the other hand, depends on the dynamics of protein–ligand interactions [3]. For example, the population of transient or “hidden”, ligand-bound states during ligand release from the protein–ligand complex can increase ligand residency times at or near the ligand-binding pocket. Ligand residency time is thought to be an important parameter in drug efficacy because a long ligand residency time provides a temporal discrimination against shorter-lived off-target interactions [3]. The importance of intermediate states can be understood in the context of the induced fit model of protein–ligand interaction [3]. In this scenario, the ligand first associates with the protein surface (as an encounter complex) to form an (or more intermediate) initial bound state(s) from where the final protein–ligand complex geometry is reached. Ligand release occurs in an opposite way to ligand binding, but requires the population of one or more intermediate bound states prior to ligand dissociation from the ligand–protein complex. At each stage of intermediate state population, the ligand can either repopulate the bound state or move toward a surface state. The determination of these intermediate states is experimentally challenging [4] because they often constitute a minor population (at steady-state equilibrium) and require specialized techniques to track them [5,6].

Red-edge excitation spectroscopy (REES) is undergoing a renaissance [7,8,9,10,11] as one method that could potentially determine the presence of minor (sub) populations in an ensemble of protein conformational ensembles. First described by Weber [12], Galley [13], and Rubinov [14], REES relies on a distribution of fluorophore-solvent interactions, referred to as in-homogenous broadening, which arises from a distribution of microenvironments around a population of fluorophores. The peak of the S_0_ → S_1_ transition in the absorption spectrum represents the most populated fluorophore-solvent microenvironment, whereas excitation at the red-edge of the absorption spectrum selectively excites less populated microenvironments where the ground state is destabilized (in energy) and the excited state is stabilized relative to the ground state. Provided the solvent environment relaxes slowly compared to the excited-state emission, then the presence of the red-edge excited species is unveiled from a characteristic red shift in the emission from the fluorophore population as the excitation wavelength is moved from the peak of the absorption spectrum to the red-edge of the absorption spectrum. If the ligand is an environment-sensitive fluorophore and the protein represents the solvent (and is static on the fluorescent timescale), then red-edge excitation spectroscopy should, in principle, be able to detect intermediate ligand bound states of ligand–protein complexes.

In the present work, our motivation was to link the red-edge excitation shift phenomenon to the properties of intermediate ligand-bound states (number, population, and energetics). To do this, we built a thermodynamic model of a fluorescent ligand interacting with a number of microstates in a protein. We used Marcus electron transfer theory [15,16] to describe the absorption and emission processes of the fluorescent ligand in each microstate. The population distribution of microstates along the protein–ligand coordinate is derived from a protein–ligand free energy landscape (Figure 1).

The rest of this paper is organized as follows. In Section 2, we describe a theory of the red-edge excitation shift for a fluorescent ligand bound to a protein. In Section 3.1, we use the model to simulate the influence of the number, population, and energetics of the ligand-bound states on the shape of the red-edge excitation shift curve. In Section 3.2, we used the model to fit the experimental red-edge excitation shift data [17] from a solvatochromic kinase inhibitor [18], under conditions of restricted protein motion [19], bound to two different enzymes [20,21]. The thermodynamic parameters derived from the model allow energetic dissection of the contribution of intermediate bound states to the ligand binding process.

## 2. Theory

### 2.1. Requirements of the Red-Edge Excitation Spectroscopy (REES) Effect

For REES to occur, there are a general set of requirements [7,11,13], which we will outline. First, the fluorophore requires a ground state and an electronic excited state, which both differ in the interaction with the environment. For this to occur, there needs to be a dipole moment difference (magnitude and/or direction) between ground and excited-states. Second, the fluorophore needs to be immersed in a range of environments, such that there is a distribution of fluorophore–environment interactions. Third, the dynamics of the environment needs to be comparable or slower than the time-scale of the fluorescence emission. We will make an additional stipulation that the dynamics of the environment is slower than the rate of fluorescence emission. This stipulation means that we only need to specify the distribution of environments in our model. We will next place these requirements in the context of a quantitative model for the REES effect.

### 2.2. Absorption and Emission from a Ligand in a Single Microstate

Consider a two-level ligand, L, with an electronic ground state, GS, and a single electronic excited-state, termed a charge-transfer or CT state. The term CT implies that the excited-state has a charge-transfer character (i.e., a dipole moment, which is different in the direction and/or magnitude to the electronic ground state), which is a requirement for the REES effect. We will treat the absorption and emission processes using classical Marcus theory such as CT absorption and CT emission, respectively.

We denote the free energy of the GS as a function of a collective coordinate *x* as *G_GS_(x)* and the corresponding free energy of the CT state, as *G_CT_(x)*, Equation (1).

From Marcus electron transfer theory [15,16], we can write the local free energies as,
(1a)$$G_{GS}\left(x\right)=\frac{kx^{2}}{2}$$
(1b)$$G_{CT}\left(x\right)=\frac{k{\left(x-x_{CT}\right)}^{2}}{2}+\Delta G_{CTGS}$$
where *k* is a constant and *x_CT_* is the displacement of the excited-state free energy surface along the coordinate *x* (with respect to the ground state free energy surface), and *∆G_CTGS_* is the vertical displacement of the minima of the ground-state and excited-state free energy surfaces. For simplicity, we have taken *k* to be equal in the GS and CT states (ignore the different impact on *k* in GS and CT states).

Assuming thermal equilibrium in the ground-state, the probability of the system being at coordinate *x*, *P(x)*, is given by the Boltzmann distribution (Equation (2)),
(2a)$$P\left(x\right)=\sqrt{2\pi Tk_{b}}e^{-(\frac{G_{GS}\left(x\right)}{k_{b}T}\;)}$$
(2b)$$P\left(x\right)=\sqrt{2\pi Tk_{b}}\;e^{-(\frac{0.5kx^{2}}{k_{b}T}\;)}$$

In Equation (2b), *P(x)* is a Gaussian function centered at *x* = 0; *k_b_* is the Boltzmann constant; and *T* is the temperature. Note that *G_GS_ (x)* is given in Equation (1a).

The absorption energy (*hν_abs_*) of a photon required to make L transition from the ground state GS to the excited state CT is given by the expression in Equation (3),
(3)$$h\nu_{abs}\left(x\right)=G_{CT}\left(x\right)-G_{GS}\left(x\right)=\;\Delta G_{CTGS}+\;\frac{k{\left(x_{CT}\right)}^{2}}{2}-kx\left(x_{CT}\right)$$

Equation (3) shows that the absorption energy is a linear function of *x*. By inference, the absorption spectrum will also be a Gaussian function of the absorption energy with a single peak (mode) at *x* = 0 and a width related to temperature and the constant *k*.

The absorption maximum is located at *x* = 0, i.e., and from Equation (4), we can see that
(4)$$h\nu_{abs}\left(x=0\right)=G_{CT}\left(0\right)-G_{GS}\left(0\right)=\;\Delta G_{CTGS}+\frac{k{\left(x_{CT}\right)}^{2}}{2}$$

The value (1/2) *k(x_CT_)^2^* is called the reorganization energy in Marcus theory.

To describe the emission process from the excited CT state to the GS ground state on the ligand L, we will assume that thermal (vibrational) relaxation occurs in the local CT state (i.e., a single microstate) prior to emission. By analogous reasoning, the emission spectrum is given as a Gaussian function with the mean (and mode) position of the excited CT state at *x* = *x_CT_*. The peak (and mean value) of emission spectrum is given by Equation (5),
(5)$$h\nu_{em}\left(x=x_{CT}\right)=G_{CT}\left(x_{CT}\right)-G_{GS}\left(x_{CT}\right)=\;\Delta G_{CTGS}-\frac{k{\left(x_{CT}\right)}^{2}}{2}$$

Combining Equations (4) and (5), we arrive at the expected results from Marcus theory, that the reorganization energy for the CT to GS transition is half the Stokes shift and that the free energy for the GS to CT transition is given as the average of the energies of the absorption and emission peaks, Equation (6a,b).
(6a)$$\frac{h\nu_{abs}\left(x=0\right)-h\nu_{em}\left(x=x_{CT}\right)}{2}=\frac{k{\left(x_{CT}\right)}^{2}}{2}$$
(6b)$$\frac{h\nu_{abs}\left(x=0\right)+h\nu_{em}\left(x=x_{CT}\right)}{2}=\;\Delta G_{CTGS}$$

The reorganization energy is normally composed of two parts, an inner-sphere contribution, which relates to changes in the geometry of the ligand L (and closely associated solvent molecules) (i.e., changes in bond lengths) and an outer-sphere contribution, which relates to changes in the solvent dipoles to accommodate the charge transfer state. As stipulated above, we will assume that the dominant contribution to relaxation in the excited CT state is due to vibrational relaxation in the ligand (i.e., inner sphere reorganization) with no or negligible reorganization of the protein or water dipoles during the excited-state lifetime.

Note that because we have assumed vibrational/inner-sphere relaxation in the local CT state, the emission will be independent of the excitation energy within the single microstate. For the single GS, single CT state considered here, the red-edge excitation shift will be zero.

### 2.3. Absorption and Emission from a Ligand in a Collection of Discrete Protein Microstates

We now consider the ligand L distributed among a collection of N discrete microstates. The motivation for considering discrete states, as opposed to a continuum, is from experimental and sophisticated computational studies, revealing the existence of discrete (yet transient) bound states of ligands in proteins [4,5,22,23,24]. For simplicity, we will assume that the microstates are equally-spaced along coordinate *x* by a displacement of *x_GS_*. We also define a protein–ligand free energy landscape FEL (i), which is a parametric function of the microstates. This definition means that for N states, we only need to define one energy parameter, as opposed to N individual parameters for the N states. We assume functions of the form *Ai^b^* where *A* and *b* are constants (*b* = 1 linear, *b* = 2 quadratic, etc.) and *i* is the *i*th microstate (starting from *i* = 0 to i = N). For example, *b* = 1 describes a free energy ramp, *b* = 2 is a quadratic free energy function.

With these prescriptions, the ground state free energy surfaces of the 0th, 1st,…, *i*th, …, and *N*th microstates are given by the expressions in Equation (7a–d).
(7a)$$G_{GS0}\left(x\right)=\frac{kx^{2}}{2}$$
(7b)$$G_{GS1}\left(x\right)=\frac{k{\left(x-x_{GS}\right)}^{2}}{2}+\;A^{}$$
(7c)$$G_{GSi}\left(x\right)=\frac{k{\left(x-ix_{GS}\right)}^{2}}{2}+\;Ai^{b}$$
(7d)$$G_{GSN}\left(x\right)=\frac{k{\left(x-Nx_{GS}\right)}^{2}}{2}+\;AN^{b}$$

We now have to consider the CT states (i.e., excited-states) of the ligand in the different microstates. We denote the free energy difference between ground-state and CT state in the 0th microstate as *∆G*_*CTGS*0_, and the corresponding quantity for the Nth microstate as *∆G_CTGSN_*. The free energy surfaces of the 0th, 1st, …, ith, …, and Nth, CT states are then given by the expressions in Equation (8a–d),
(8a)$$G_{CT0}\left(x\right)=\frac{k{\left(x-x_{CT}\right)}^{2}}{2}+\;\Delta G_{CTGS0}$$
(8b)$$G_{CT1}\left(x\right)=\frac{k{\left(x-x_{GS}-x_{CT}\right)}^{2}}{2}+\;A^{}+\;\Delta G_{CTGS0}\;+(\frac{1}{N})\left(\Delta G_{CTGSN}-\;\Delta G_{CTGS0}\right)$$
(8c)$$G_{CTi}\left(x\right)=\frac{k{\left(x-ix_{GS}-x_{CT}\right)}^{2}}{2}+\;Ai^{b}+\;\Delta G_{CTGS0}\;+(\frac{i}{N})\left(\Delta G_{CTGSN}-\;\Delta G_{CTGS0}\right)$$
(8d)$$G_{CTN}\left(x\right)=\frac{k{\left(x-Nx_{GS}-x_{CT}\right)}^{2}}{2}+\;AN^{b}+\;\Delta G_{CTGSN}$$

In Equations (9a–d), we have implicitly assumed that the reorganization energy for the CT to GS transition is independent of the microstate. This assumption is in accordance with the stipulation of limited outer-sphere reorganization of the protein environment during the excited-state lifetime of the ligand (i.e., only vibrational relaxation of the ligand contributes to the Stokes shift). Note that in the construction of the CT surfaces of the microstates, three additional parameters are needed: *x_CT_*, *∆G*_*CTGS*0_, and *∆G_CTGSN_*. We will see later that these quantities can be determined from experiment.

We can immediately distinguish different behaviors of the microstates depending on the different parameter values characterizing the FEL.

(i) When *∆G_CTGSN_* < *∆G*_*CTGS*0_ and A > 0, b > 1, as we proceed along the FEL(i) in the 0 to N direction, the ground state increases in energy, while the CT states become stabilized in energy with respect to the GS. This is the qualitative behavior of the red-edge excitation effect (i.e., red-edge excitation leading to a red-shift in the emission).

(ii) When *∆G_CTGSN_* > *∆G*_*CTGS*0_ and A > 0, b > 1, as we proceed along the FEL(i) in the 0 to N direction, the ground state increases in energy, while the CT states become destabilized in energy with respect to the GS. This is the qualitative behavior of the red-edge excitation, leading to increasing blue shifted emission (i.e., an unorthodox red-edge effect).

(iii) Combinations of (i) and (ii) can lead to essentially either a red-edge effect or blue emission upon increasing red-excitation.

Having defined the FEL, we now consider the behavior of the ligand L within the ensemble of the N microstates (i.e., at thermal equilibrium). The population distribution of ligand L, among the microstates, depends on the FEL and temperature via the Boltzmann equation. The populations of state 0, 1, i, and N are given by the expressions in Equation (9a–d),
(9a)$$P_{GS0}\left(x\right)=e^{-(\frac{G_{GS0}\left(x\right)}{k_{b}T}\;)}$$
(9b)$$P_{GS1}\left(x\right)=e^{-(\frac{G_{GS1}\left(x\right)}{k_{b}T}\;)}$$
(9c)$$P_{GSi}\left(x\right)=e^{-(\frac{G_{GSi}\left(x\right)}{k_{b}T}\;)}$$
(9d)$$P_{GSN}\left(x\right)=e^{-(\frac{G_{GSN}\left(x\right)}{k_{b}T}\;)}$$

The relative weights of microstates 0, 1,…, *i*,… and *N* are defined in Equations (10a–e),
(10a)$$W_{GS0}\left(x\right)=P_{GS0}\left(x\right)/Z\left(x\right)$$
(10b)$$W_{GS1}\left(x\right)=P_{GS1}\left(x\right)/Z\left(x\right)$$
(10c)$$W_{GSi}\left(x\right)=P_{GSi}\left(x\right)/Z\left(x\right)$$
(10d)$$W_{GSN}\left(x\right)=P_{GSN}\left(x\right)/Z\left(x\right)$$
(10e)$$Z\left(x\right)=P_{GS0}\left(x\right)+P_{GS1}\left(x\right)+P_{GSi}\left(x\right)+\cdots +P_{GSN}\left(x\right)$$

It is useful to define the absorbance energy for the GS to CT transition of each microstate. The absorbance energy for each microstate 0, 1, …, *i*, …, and *N* are denoted by Equation (11a–d),
(11a)$$h\nu_{abs0}\left(x\right)=\;\Delta G_{CTGS0}+\frac{k{\left(x_{CT}\right)}^{2}}{2}-kx\left(x_{CT}\right)$$
(11b)$$h\nu_{abs1}\left(x\right)=(\frac{1}{N})\left(\Delta G_{CTGSN}-\;\Delta G_{CTGS0}\right)+\;\Delta G_{CTGS0}+\frac{k{\left(x_{CT}\right)}^{2}}{2}-kx\left(x_{CT}\right)$$
(11c)$$h\nu_{absi}\left(x\right)=(\frac{i}{N})\left(\Delta G_{CTGSN}-\;\Delta G_{CTGS0}\right)+\;\Delta G_{CTGS0}+\frac{k{\left(x_{CT}\right)}^{2}}{2}-kx\left(x_{CT}\right)$$
(11d)$$h\nu_{absN}\left(x\right)=\Delta G_{CTGSN}+\frac{k{\left(x_{CT}\right)}^{2}}{2}-kx\left(x_{CT}\right)$$

The emission from each microstate is considered to occur from the equilibrated energy minimum of the CT state within each microstate, but not between microstates (no transfer between microstates during the excited-state lifetime). The position of the emission peak (and average emission energy) of each microstate is given by Equation (12a–d),
(12a)$$h\nu_{em0}\left(x\right)=\;\Delta G_{CTGS0}-\frac{k{\left(x_{CT}\right)}^{2}}{2}-kx\left(x_{CT}\right)$$
(12b)$$h\nu_{em1}\left(x\right)=\left(\frac{1}{N}\right)\left(\Delta G_{CTGSN}-\;\Delta G_{CTGS0}\right)+\;\Delta G_{CTGS0}-\frac{k{\left(x_{CT}\right)}^{2}}{2}-kx\left(x_{CT}\right)$$
(12c)$$h\nu_{emi}\left(x\right)=\left(\frac{i}{N}\right)\left(\Delta G_{CTGSN}-\;\Delta G_{CTGS0}\right)+\;\Delta G_{CTGS0}-\frac{k{\left(x_{CT}\right)}^{2}}{2}-kx\left(x_{CT}\right)$$
(12d)$$h\nu_{emN}\left(x\right)=\Delta G_{CTGSN}-\frac{k{\left(x_{CT}\right)}^{2}}{2}-kx\left(x_{CT}\right)$$

We now turn to the determination of how the ensemble behaves in photon absorption and emission.

The ensemble absorption energy along coordinate x is given as the weighted sum of the absorbances of the individual microstates (Equation (13)),
(13)$$h\nu_{abstot}\left(x\right)=h\nu_{abs0}\left(x\right)W_{GS0}\left(x\right)+h\nu_{abs1}\left(x\right)W_{GS1}\left(x\right)+\cdots +h\nu_{absN}\left(x\right)W_{GSN}\left(x\right)$$

The average emission energy from the ensemble along coordinate x is given as the weighted sum of the average emissions from the individual microstates (Equation (14)),
(14)$$h\nu_{emtot}\left(x\right)=h\nu_{em0}W_{GS0}\left(x\right)+h\nu_{em1}W_{GS1}\left(x\right)+\cdots +h\nu_{emN}W_{GSN}\left(x\right)$$

A plot of the LHS of Equation (14) as a function of LHS of Equation (13) is the average emission energy as a function of the absorbance energy (i.e., the red-edge excitation shift in energy space). The REES curve depends on the free energy landscape parameters (A and b), the microstate spacing (*x_GS_*), the number of microstates N, and L parameters (reorganization energy ((1/2))*κx_CT_^2^*), and min and max free energy gaps (*∆G_CTGSN_,∆G_CTGS0_*).

The photo-physical parameters pertaining to the ligand L dictate the separation in energy between the highest energy blue emission and the lowest energy red emission as well as the peak (blue) absorption and red absorption. The parameters thus dictate the positions and ranges of observables, but do control the shape of the REES curve. The reorganization energy and free energy gaps can be obtained from spectral data provided the peak absorbance and blue/red emissions can be isolated.

From the absorbance and emission energies of the highest energy or lowest energy states, the reorganization energy can be obtained from half the Stokes shift and the free energy gaps obtained from the average of the absorbance and emission energies (Equations (15a–d)),
(15a)$$\frac{k{\left(x_{CT}\right)}^{2}}{2}=\frac{h\nu_{absN}\left(max\right)-h\nu_{emN}\left(\max\right)}{2}=\frac{h\nu_{abs0}\left(max\right)-h\nu_{em0}\left(\max\right)}{2}$$
(15b)$$\Delta G_{CTGS0}=\frac{h\nu_{abs0}\left(max\right)+h\nu_{em0}\left(\max\right)}{2}$$
(15c)$$\Delta G_{CTGSN}=\frac{h\nu_{absN}\left(max\right)+h\nu_{emN}\left(\max\right)}{2}$$
(15d)$$\left(\Delta G_{CTGS0}-\;\Delta G_{CTGSN}\right)=h\nu_{em0}\left(max\right)-h\nu_{emN}\left(\max\right)$$

In practice, one would normally assign *hν_abs0_* (max) from the S_0_ → S_1_ absorbance peak of the ligand in the ligand–protein complex (or possibly the equivalent quantity from the fluorescence excitation spectrum) and *hν_em0_* from the corresponding emission peak. *hν_emN_* could be inferred from the emission peak upon red-excitation or alternatively from data fitting. We will discuss the modeling in the next section.

Naturally, the parameters that control the energies and positions of the microstates do influence the shape of the REES curve. To show this, we will simulate different conditions that may occur in actual ligand–protein-complexes. These simulations are presented in Section 3.1.

### 2.4. Modeling Approach

Equations (1)–(15) present the mathematical description of the model at hand and at first sight contain many parameters. In this sub-section, we explain our approach to fitting a dataset, how we obtained parameters from experiment, and how we evaluated and justified different models. As described previously [17], we found that a sigmoidal fit was sufficient to describe REES data from kinase inhibitor–kinase complexes. This model contains four parameters, two which control the end-points of the curve and two parameters that control the mid-point and the slope of the mid-point in the transition region. This sigmoidal curve provides a mathematical representation of the REES data, which is used in subsequent model fitting and interpretation. The parameters in the REES model for the ligand can be obtained from the end-points of the sigmoidal fit to the REES data and either the absorption maximum (or the excitation spectrum maximum). Specifically, from Equation (15b), *∆G_CTS0_* can be obtained from absorbance maximum and the blue-emission end-point in the REES curve, while *∆G_CTSN_* can obtained from the blue and red endpoints and *∆G_CTS0_*. The reorganization energy (and thus *x_CT_*) can be obtained from the absorbance maximum and the blue end-point from the sigmoidal fit. Thus, while there are three parameters associated with the ligand, all of them can be obtained from the experiment. To model the REES data using the FEL model, we first modeled the REES data with one intermediate (i.e., N = 1). This model has four free energy surfaces, one pair (ground and excited state) for the bound state and one pair (ground and excited state) for the intermediate state. This model has been published previously to describe red-edge effects in other biological macromolecules [25]. This model contains the least number of parameters (three parameters for the ligand and two parameters for the intermediate state). Note that because the ligand parameters were fixed from the experiment, the number of parameters needed to fit the REES transition region for this model was two. We performed this fit minimizing the sum least squares from the sigmoid fit and the theoretical model.

The next level of complexity involves models with two or more intermediate states. We created models with N = 2 up to 20 intermediate states. Each model was a separate model (in the same way that a fit of fluorescence decay curve to one exponential decay is different from a model that is a fit to two exponential decays). These models contained two extra parameters that define the FEL of the microstates (compared to the single intermediate model), these parameters are N (the number of intermediate states) and b. We found that b = 2 worked best and this agreed well with the free energy functional from a recent study of a kinase inhibitor–kinase complex [4]. Quadratic free energy functionals abound in biology, physics, and chemistry and for the present model, imply that the probability of the N^th^ state has a Gaussian probably function with N. Nevertheless, b and N were both varied to obtain an optimum fit.

The increase in the number of parameters with the N > 1 models needs to be justified on the basis of some criterion. We used the Akaike information criterion (AIC), which allows a comparison of models with different numbers of parameters and balances the need for good data fitting with over-parameterization. The formula for the AIC value is given in Equation (16),
(16)AIC=2k+nlnLSQ
where *k* is the number of parameters; *n* is the number of datapoints; *LSQ* is the sum of squared residuals; and *LN* is the natural logarithm. For example, for the p38a-inhibitor complex, with N = 1, LSQ = 10, and with N = 10, LSQ = 4. With 10 datapoints, nLog(LSQ) = 23 (N = 1 model) and nlog(LSQ) = 13.8 (n = 10), which was a larger difference than 2k = 4 (penalty for the greater number of parameters) with the N = 10 model.

## 3. Results

### 3.1. Simulations of REES Curves for Different Protein–Ligand Free Energy Landscapes

To understand how the theoretical REES curve depends on parameters of the protein–ligand free energy landscape, we have carried out simulations for a defined system consisting of 10 ligand–protein microstates. The spectroscopic parameters of the ligand L were set to (*x_CT_* = 0.89 (eV)^0.5^, *κ* = 1, *∆G*_*CTGS*0_ = 3.346 eV, *∆G_CTGSN_* = 3.223 eV), which corresponds to an emission peak in the range of 420nm to 438nm (note that these values are somewhat arbitrary).

Figure 2 contains a series of REES curves as a function of the energy parameters of the protein–ligand free energy landscape. It is notable that these curves are sigmoidal functions with emission wavelength limits matching those input into the model, as expected. In Figure 2A, *x_GS_* was fixed and b was set to 1 (b = 1 (linear ramp) and A was variable. This defines the FEL as a linear ramp function. As can be seen in Figure 2A, increasing the value of the energy parameter A (i.e., the energy gradient), causes a significant change in the position of the mid-point of the sigmodal REES function along the excitation wavelength axis (i.e., to longer wavelengths). In Figure 2B, A and b were fixed, and the influence on the spacing of the free energy surfaces of the microstates along coordinate *x* (i.e., the parameter *x_GS_*) was determined. Increasing the value of *x_GS_* changed the shape of the REES curve, causing the gradient near the mid-point to increase significantly.

In the linear ramp model for the free energy landscape (b = 1), the free energy difference between the successive minima of the microstate free energy surfaces is given by Equation (17),
(17)$$G_{GS1}\left(min\right)-G_{GS0}\left(min\right)=\Delta G_{GSi+1,i}=A$$

And the ground-state reorganization energy, RE (Equation (18)) is
(18)$$\mathrm{RE}=\frac{k{\left(x_{GS}\right)}^{2}}{2}$$

From Marcus theory, the activation energy between microstates is (in the linear ramp model) is given by Equation (19),
(19)$$\Delta G_{GSi+1,i}act=\frac{{\left(\Delta G_{GSi+1,i}+\mathrm{RE}\right)}^{2}}{4RE}$$

In the context of a linear protein–ligand FEL, shifts in the mid-point of the REES curve indicate changes in the relative thermodynamic stability of the most populated state relative to the least populated state. Changes in the steepness of the REES transition (all other things being equal) would indicate a change in the roughness of the FEL (i.e., increase or decrease in activation energy between successive microstates). A rougher FEL produces a steeper REES transition.

We also considered more complex protein–ligand energy landscapes such as quadratic (or harmonic) free energy as a function of the microstate. The results of a simulation on the influence of changing the A parameter on the REES curves are shown in Figure 3 for the quadratic FEL (i.e., the ith microstate free energy proportional to i^2^). The simulations reveal that increasing A (which is analogous to a Hook’s law spring constant) causes both a shift in the mid-point of the REES curve (along the excitation wavelength axis to longer wavelengths) and a consequential change in the steepness of the REES transition (more shallow appearing transition). In Figure 3B, A and b were fixed, and the influence on the spacing of the free energy surfaces of the microstates along coordinate x (i.e., the parameter *x_GS_*) was determined. Increasing the value of *x_GS_*, changes the shape of the REES curve, causing the gradient near the mid-point to increase significantly.

If we examine the free energy landscape for the quadratic model (b = 2), we note that the free energy difference between the successive minima of the microstate free energy surfaces is (Equation (20)),
(20)$$G_{GSi}\left(min\right)-G_{GSi+1}\left(min\right)=\Delta G_{GSi+1,i}=A\left(2i+1\right)$$

From Marcus theory, the activation energy between successive microstates is (in the quadratic FEL model) given by Equation (21),
(21)$$\Delta G_{GSi+1,i}act=\frac{{\left(A\left(2i+1\right)+\mathrm{RE}\right)}^{2}}{4RE}$$

By inspection of Equation (20), we see that in the quadratic FEL model, the successive free energy differences between the microstate minimum increase with the *i*th microstate, and consequently the Marcus activation energy between microstates is also no longer independent of the microstate (Equation (21)). According to the quadratic FEL model (Equation (21)), the activation barrier between successive microstates increases with increasing i, while the activation barriers decrease in the opposite direction.

To explore the effect of different numbers of microstates on the REES curves, we fixed all the parameters in the protein–ligand FEL and ligand L (parameters were b = 2, A = 0.0025, *x_GS_* = 0.15)) and varied N from 5, 10, 15, and 20 microstates. The influence of the number of microstates, N, on the REES curves is depicted in Figure 4. An increase in the number of microstates, N, is accompanied by a substantial change in shape of the REES curves. In general, both the position of the mid-point and the slope of the curve at the mid-point changed with N. Increasing N decreased the slope near the transition mid-point and increased the excitation wavelength at which the mid-point transition occurred.

The parameters chosen in these simulations were in the range of what we might expect for intermediate states in protein–ligand complexes. The free energy difference between the most stable state in the protein–ligand complex and the least stable bound state in the protein ligand complex was 10A (for N = 10 and for the linear model, A = 0.0025–0.05 eV). This corresponds to a free energy value range from 0.025 to 0.5 eV or 2.5–50 KJ/mol or about 0.6–12 kcal/mol (2 *k_b_T* to 20 *k_b_T* (k_b_ is Boltzmann constant)). By way of comparison, the total free energy between the lowest energy bound state and a totally-free ligand state between 6.9 *k_b_T* and 20.7 *k_b_T* corresponds to an equilibrium dissociation constant of between 1 mM and 1 nM (i.e., the range of equilibrium constants for the majority of protein–ligand interactions).

### 3.2. Application to REES Data from Kinase Inhibitor–Kinase Complexes

To provide a concrete example of the application of the theoretical models to real data, we refer to recent published data from the anilino-quinazoline small molecule kinase inhibitor (AG1478) in complex with a kinase (MAPK14 or p38α kinase) [17]. AG1478 is an intrinsically-fluorescent kinase inhibitor and exhibits environmentally-sensitive fluorescence with a Stokes shift as large as 100 nm in highly polar solvents [18]. Therefore, AG1478 is a suitable ligand for REES studies. Spectroscopic studies on the AG1478-p38α kinase complex revealed a progressive red-shift in the emission spectrum with increasing excitation wavelength, consistent with a REES-type effect. The magnitude of the total red-edge shift was also largely independent of temperature (in the range 283–313 K) [17], which is an indication [19] that the motions of the environment are restricted on the nanosecond time-scale. In this case, we can apply the theory outlined here to analyze the REES data. Figure 5 depicts the REES curve as a solid line represented by the Boltzmann sigmoid fitting function used in the publication [17]. Individual datapoints corresponding to the REES data are indicated by the open circles in the inset to Figure 5. To fit this data to our thermodynamic model, we first optimized the photo-physical parameters of the AG1478 ligand in our model to provide a good fit to the extremes of the REES curve (i.e., to match the initial and final emission values in the REES curve). In our analysis, we then fixed the values for the ligand photo-physics (see Table 1 for parameter values). We then chose the form of the FEL function (linear or quadratic or cubic) and then performed a least-squares fit to the experimental curve by adjusting two parameters, the energy parameter A and the microstate spacing parameter *x_GS_*. The number of microstates was initially fixed to N = 10 per simulations in the previous section. Table 2 outlines the results of each analysis, together with the sum of squares for the fit. Plots of the simulated REES curves are shown in the inset of Figure 5 to afford visual comparison. The linear model provided a reasonable fit with a SLS value of 53.5. The cubic model also provided a reasonable fit to the data with a SLS value of 204. However, the quadratic model outperformed both models with a SLS value of 3.5 (i.e., by a factor of 10 improvement). All models had the same number of adjustable parameters; therefore, we rejected the linear and cubic models, in favor of the quadratic model.

In fitting the REES curve for the AG1478-p38α kinase complex, we initially assumed that the number of microstates was N = 10. We repeated the analysis of the REES curve for different values of N for the quadratic FEL model as described above, (range of N = 3 to 20). The goodness-of-fit parameter as a function of the number of microstates is depicted in Figure 6A. As can be seen in Figure 6A, the goodness-of-fit is dependent on the number of microstates in the model, with N = 10 providing the best fit to the data.

An important concern is whether by pre-selecting a quadratic FEL, we are biasing the fitting in some way. Therefore, we repeated the analysis of the p38α-kinase-AG1478 REES data, allowing all the parameters of the FEL to vary. The goodness-of-fit parameter versus number of microstates revealed a clear minimum (near N = 9–10) with the FEL exponent close to 2 (b = 1.94). To examine the robustness of our fitting procedures and model in more detail, we simulated REES data, fit the simulated REES curve to a sigmoidal function (as we do for experimental data), then fit the resulting sigmoid to a FEL model. Table 3 compares the input parameters with output parameters for FELs over an energy range of 0.04–0.24 eV. It is noted that the output parameters were within 15% of the input parameters.

A plot of the free energy landscape of the protein–ligand complex AG1478-p38α kinase is shown in Figure 7A, depicting the relative free energies of the 10 protein–ligand microstates as a function of the generalized protein–ligand coordinate. According to the quadratic FEL model and the fit parameters, the free energy difference between the most stable bound state and the least stable bound state in the AG1478-p38α kinase complex was −0.144 eV (or −5.6 *k_b_T* or −14.4 kJ/mol or −3.5 kcal/mol). The reported inhibition constant for the AG1478-p38α kinase complex was 0.560 µM (*20*). A 0.6 µM equilibrium dissociation constant corresponded to a free energy difference between bound and free AG1478 of −14.4 *k_b_T*. According to our analysis, the intermediate states of AG1478-p38α kinase could contribute as much as ~40% (= 5.6/14.4) to the total free energy of binding.

To test this idea further, we also analyzed the published REES data from AG1478 bound to another kinase, aminoglycophosphokinase (APH) [17] which is a threonine-serine kinase derived from bacteria, but has a similar fold to eukaryotic protein kinases. A plot of the REES curve for the AG1478-APH complex is depicted in Figure 5B, REES data in the Figure 5B inset, photophysical parameters in Table 4, and analysis of different models collected in Table 5. We found the best fit was to a quadratic model (Table 4) for the FEL surface with five microstates (i.e., N = 5, Figure 6B). From the model fit, the free energy difference between the most stable state and the least stable state in the AG1478-APH kinase complex was −0.03 eV or approximately −1 *k_b_T*.

A comparison of the free energy landscapes for the two kinase-AG1478 complexes is shown in Figure 7 (generated using Equation (7)). It is notable that as a function of the protein–ligand reaction coordinate (x in Equations (1)–(14)), the p38α-AG1478 free energy profile was steeper while the APH-AG1478 the free energy profile was shallower. From a physics analogy of a Harmonic spring, one would deduce that it is harder (i.e., costs more energy) to displace AG1478 from its most stable bound state in the p38α kinase complex along the reaction coordinate (to its least stable bound state) than to displace the most-stable bound state in the corresponding APH-AG1478 complex because the AG1478-p38α kinase complex interaction has a larger effective spring constant. This qualitatively agrees with the relative inhibition constants of the two proteins. The inhibition dissociation constant for p38α kinase-AG1478 was K_i_(p38α kinase) = 0.560 µM and represents “tighter” binding than for APH-AG1478 interaction, with an inhibition dissociation constant of K_i_(APH) = 15.6 µM (*21*).

At equilibrium, the total binding free energy for the protein–ligand interaction (∆G_protein–ligand_) is the sum of the free energy for ligand association with protein (∆G_assoc_) to a surface state (or encounter complex) and a free energy for ligand intrusion (∆G_intrusion_) from a surface-bound state to the interior-bound state. If we equate (∆G_intrusion_) with the free energy difference between the most-stable bound state and the least-stable bound state, then this quantity may be extracted from the protein–ligand FELs derived from REES data. From the FELs for the AG1478 complexes with APH and p38α kinase, we estimated (∆G_intrusion_)_APH-AG1478_ = −0.7 kcal/mol and (∆G_intrusion_)_p38α-G1478_ = −3.3 kcal/mol. The total free energies for binding (from the inhibition constants) were (∆G_protein-ligand_)_APH-AG1478_ = −6.5 kcal/mol for the APH-AG1478 interaction and (∆G_protein–ligand_)_p38 α−AG1478_ = −8.3 kcal/mol for the p38α-AG1478 interaction. From ∆G_intrusion_ and ∆G_protein–ligand_, we estimated (∆G_assoc_)_APH-AG1478_ = −5.8 kcal/mol and (∆G_assoc_)_p38a-AG1478_ = −5.0 kcal/mol. Comparing the differences between the two proteins, we can see that the difference in total free energies for binding ((δ∆G_protein-ligand_)_p38a/APH_ = 1.84 kcal/mol) was more than accounted for by the difference in the free energies for intrusion ((δ∆G_intrusion_)_p38a/APH_ = 2.6 kcal/mol). In contrast, the difference in free energies for surface association ((δ∆G_assoc_)_p38a/APH_ = −0.8 kcal/mol) makes a comparatively smaller contribution. Thus, the energetics of the ligand-bound states of the protein–ligand complexes is a differentiating factor in explaining the different inhibition constants of AG1478 for two different kinases.

The protein–ligand free energy landscapes, derived from the REES data, also provide estimates of activation barriers for transitions between the different ligand–protein microstates (along the reaction coordinate, x). As eluded to above, for the quadratic FEL, the activation barrier is related to the free energy difference and the reorganization energy between microstates (see Equation 20). For the APH-AG1478 complex, the largest activation barrier between the 4th and 5th microstate was 0.016 eV. For the p38α kinase-AG1478 complex, the largest activation barrier between successive microstates was between the 9th and 10th microstates and was 0.031 eV. Thus for both complexes, the successive barrier heights were not significantly greater than the thermal energy (i.e., *k_b_T* = 0.025 eV at 298K). From the model, one would also predict that the total magnitude of the REES effect is not very dependent on temperature, which agrees qualitatively with published observations over the temperature range 283–308 K [17].

Our focus thus far has been on the properties of the FEL (i.e., in the electronic ground states of the ligand in the microstates defined by the protein–ligand interaction). The extrema of the REES curves and associated analysis provide useful information on the CT states, and by inference, on the types of environments encountered by the ligand in the protein microstates (i.e., polarity of the amino acids and/or relative hydration). The relative displacement of the CT state relative to the GS free energy surfaces, *x_CT_*, indicates the change in structure of the ligand (and possibly environment) between the GS and CT states. From this displacement *x_CT_*, we can determine the Marcus reorganization energy for the CT to GS transition, equivalent to half the Stokes shift. Comparing the *x_CT_* values for the two protein–ligand complexes (c/f Table 1 and Table 3), the displacement associated with the CT–GS transition in AG1478-APH (*x_CT_* = 1.1) was slightly larger than the corresponding quantity in the AG1478-p38α complex (*x_CT_* = 0.9). Consequentially, the reorganization energy for the CT-GS transition in AG1478-APH was 0.6 eV and for AG1478-p38a, it was 0.4 eV. The reorganization energy consists of two parts, the inner-sphere contribution (from changes in the geometry of the AG1478 and tightly-associated molecules), and an outer-sphere part, due to changes in dipolar reorganization of the environment around the AG1478. The inner-sphere term can be determined from the fluorescence of AG1478 in a non-polar solvent (where the outer-sphere contribution due to solvent dipolar reorganization is zero). From the published emission of AG1478 in non-polar solvents (toluene and dioxane), we estimated the inner-sphere reorganization energy in AG1478 to be 0.4 eV [18]. Thus, in the AG1478–p38α kinase complex and the AG1478–APH complex, the inner-sphere reorganization of the ligand is the dominant contribution to the reorganization energy. This suggests that the dipolar reorganization of the protein environment during the excited state lifetime is restricted in the AG1478-APH complex and absent (or undetected) in the AG1478-p38α kinase complexes. The free energy gap between CT and GS provide valuable information on the stabilization of the CT state relative to the GS by the protein environment and water. In the AG1478–APH complex, the CT states were more stabilized with respect to the GS (*∆G*_*CTGS*0_ = 3.18 eV, *∆G_CTGSN_* = 3.02 eV) than the corresponding AG1478–p38α kinase complex (*∆G*_*CTGS*0_ = 3.35 eV, *∆G_CTGSN_* = 3.22 eV). These observations suggest that AG1478 experiences a more polar environment along its microstate trajectory in the APH protein compared to the p38α kinase protein. It is tempting to speculate that the higher affinity of AG1478 for p38α than APH is due to the more hydrophobic binding environments in p38α, perhaps contributed in part by the nature and number of water molecules and the amino acids comprising the ligand binding pocket. However, the polarity gradients along the microstate trajectories (*∆G_CTGSN_* − *∆G*_*CTGS*0_ = −0.16eV (APH); *∆G_CTGSN_* − *∆G*_*CTGS*0_ = −0.13eV (p38α kinase)) were similar in direction and magnitude in both proteins. This may imply that least populated ligand states are more hydrated than the most populated ligand state, a view that is congruent with the idea that ligands become more hydrated as they leave the protein.

## 4. Discussion

The goal of this work was to link the properties of the ligand–protein free energy landscape to spectroscopic properties of a fluorescent ligand, namely the red-edge excitation shift. Our model enabled us to determine the number and energetics (i.e., populations) of intermediate bound states as well as the associated environments.

There are several assumptions underlying this work. First, we assumed that a thermodynamic model would provide an appropriate framework to link spectroscopy with populations and energetics. The assumption holds, provided that protein–ligand intermediates (microstates) are at thermal equilibrium in the ground electronic state. Spectroscopic measurements of proteins and ligands in closed cuvettes and under constant temperature regulation ensure that any changes in temperature or concentration of reagents are too small during the total measurement time to invalidate this assumption.

The second assumption is that each microstate retains its identity during the process of photon absorption and fluorescence. In other words, exchange between microstates is slower than the rate of fluorescence emission. Dynamic averaging between microstates would lead to a reduction in the REES effect, a result that would underestimate the number of distinct states and invalidate the derived thermodynamic parameters. Dynamic averaging can be evaluated by changing the lifetime of the probe or changing the dynamics of the protein. For example, a decrease in the red-edge effect upon heating indicates that environmental motions are on a time scale comparable to fluorescence. A temperature-independent REES (provided the protein is still in its native state) is one hallmark of restricted environment motion on the fluorescence timescale [19]. For the ligand–protein complexes studied in this paper, REES was found to be temperature independent and so this assumption likely holds in the present case. For cases where dynamic averaging is suspected, a different approach to the one posited here is required.

Our model for the free energy landscape emphasizes a small (i.e., 5–20 microstates) number of discrete protein–ligand microstates. This model is in accordance with recent molecular dynamics simulations showing hidden poses of ligands bound at various locations to a protein, along a protein–ligand binding trajectory. For the interaction of the PP1 drug with c-Src kinase, the Sugita lab identified one distinct fully-bound ligand pose as well as four semi-bound, four intermediate, and four encounter complexes [4]. While for the release of Dasatinib from c-Src, the Berne laboratory identified six states corresponding to five ligand-bound poses as well as an unbound state [21]. The number of intermediate bound states for AG1478 with APH was found to be five in the current work, all very close in energy (i.e., within thermal energy) to the most stable bound state. This analysis compared remarkably well with the number of ligand poses described in the x-ray derived structure of the APH–AG1478 complex. Between 9–10 bound states were found for AG1478 with p38α kinase from our model. Docking studies with a hydrophobic ligand found 19 potential binding poses in p38α kinase [23]. We wish to stress that the number of intermediates obtained is model-dependent (not absolute) and in our model, it is limited by the number of spectroscopically-distinguishable classes of states that we can determine. Thus the number of intermediate states extracted from the analysis here may be different to those reported by other means.

One possible interpretation of our results is that as the ligand proceeds along the protein–ligand trajectory away from the most stable binding configuration toward the protein surface, it encounters more polar or hydrophilic environments that stabilize the CT state with respect to the GS state. Our reaction coordinate then might correspond to the (relative) displacement in the center of masses of the protein and the ligand. This coincides with the view that ligands become more hydrated as they gradually leave the protein. However, our approach did not allow us to determine the kinetic path of ligand binding (or release), only the relative energetics and environments of these different states.

The free energy values obtained in the current study can be compared with values obtained by other labs for kinase–drug interactions. For the interaction of a PP1 drug with c-Src kinase (IC_50_ = 0.17µM), the Sugita lab [4] used molecular dynamics simulations and arrived at a value for the free energy for intrusion of ∆G_intrusion_ = −4.1 kcal/mol. This value is similar to the experimental value obtained in the present study of ∆G_intrusion_ = −3.5 kcal/mol for the p38α–AG1478 complex, which also has a sub-micromolar inhibition constant. Sugita estimated the free energy change associated with the encounter complex step to be ∆G_association_ = −5 kcal/mol for PP1-Src kinase, which is also close to our estimated value of (∆G_assoc_)_p38a-AG1478_ = −5.0 kcal/mol for the p38α–AG1478 association. The Kern lab [24] has revealed that the physical association step of the drug Glevec with Abl is associated with a free energy change of about −6.5 kcal/mol, while there was an additional free energy of intrusion (associated with a conformational change) step of −4.5 kcal/mol. Collectively, these results suggest that the free energy for encounter complex formation accounts for about −5–−6.5 kcal/mol of the total binding free energy. Thus, high affinity binding, in these cases at least, is largely dictated by additional steps after encounter-complex formation.

The quadratic model was the preferred FEL model (compared with linear or cubic) for the two protein–ligand complexes tested (free floating the FEL exponent also resulted in values close to two). The quadratic model also agreed quite well with the published free energy landscape derived from the molecular dynamics simulations for PP1 and c-Src kinase [4] near the minimum of the free energy landscape and in the region of the bound and semi-bound ligand poses. Free energy functions derived from molecular dynamics studies revealed that the free energy was approximately quadratic as a function of the radial distance of the drug from the binding site [4]. Since we expect microstates experiencing more polar environments to be further from the binding site toward the protein surface, a quadratic dependence of the free energy on microstate number N seems a reasonable first order approximation. Note that we have attempted more complex free energy functionals such as the Morse potential function, but these have more parameters with no improvement in the fit quality. Our mathematical model for the free energy landscape function-linear, quadratic-, or cubic- is of course somewhat simplistic. Real free energy landscapes are complex and can be very bumpy, containing many hills and valleys. As we learn about real protein–ligand FELs, more complex functions may be used to describe the energetics of the protein–ligand microstates and can be easily incorporated into the current model.

One of the main results of this paper is to provide a theoretical shape for the REES effect curve under different idealized conditions. Our simulations suggest that the full REES curve is a sigmoidal-like function. This is expected as binding models derived from thermodynamic considerations for many states are also often sigmoidal or Hill-type functions [26]. In the REES experiment, changing the excitation wavelength to longer values is similar to adding a ligand (or denaturant) to a protein (i.e., shifting the (observed) population distribution). An important difference with REES is that the population of states is not perturbed during the experiment, only the observation window. However, there are conditions where a full sigmodal curve may not be obtained from a REES experiment and only some part of a sigmoidal curve may be detected. Thus, an apparent linear, exponential, or Gaussian REES curve may be measured under certain circumstances. The model presented here allows simulation of under what conditions these different types of REES curves may be observed. However, we caution that extracting parameters from REES curves that are not strictly sigmoidal may be problematic (i.e., solving the inverse problem in this case). This issue is similar to trying to find binding constants or denaturation free energies from incomplete binding or denaturation data.

Perhaps the main virtue of the approach presented here is to allow for a comparison of REES curves from different protein–ligand complexes or under different conditions to be placed under the same conceptual framework. For example, applying the model to real REES data allowed us to compare the properties and stabilities of the intermediate ligand-bound states in two different proteins bound to an intrinsically fluorescent small molecule inhibitor. Encouragingly, the relative energetics deduced from our model correlated well with previously published relative inhibition constants as well as related thermodynamic data from other kinase–inhibitor complexes.

The REES approach may not necessarily be restricted to proteins with well-defined tertiary folds and secondary structural elements. REES is an attractive approach for examining dynamics and structure of disordered states, molten globule states, and intrinsically-disordered proteins, which may exhibit multiple conformational states along a rich conformational protein landscape [7,8,10].

## 5. Conclusions

The present statistical mechanical model allows for the interpretation of red-edge excitation shift spectroscopy (REES) curves from protein–ligand complexes under different conditions. The approach leads to a novel way to study ligand–protein binding from the determination of less populated, intermediate, and transient bound states of protein–ligand complexes, which has been experimentally challenging. The model revealed the relationship between the progressive emission red shift with red-edge excitation and the protein–ligand free energy landscape, and relates thermodynamic and photo-physical parameters to the fluorescent ligand. We applied the theoretical model to available red-edge excitation shift data from the small molecule inhibitor–kinase complexes AG1478–APH and AG1478–p38α. It was discovered that a quadratic protein–ligand free energy landscape model provided a good description of the data from both inhibitor–kinase complexes. The derived thermodynamic parameters allowed dissection of the energetic contribution of intermediate bound states to inhibitor–kinase interactions. A further study using quantum mechanical calculations in this direction is under investigation.

## Figures and Tables

**Figure 1 ijms-22-02582-f001:**
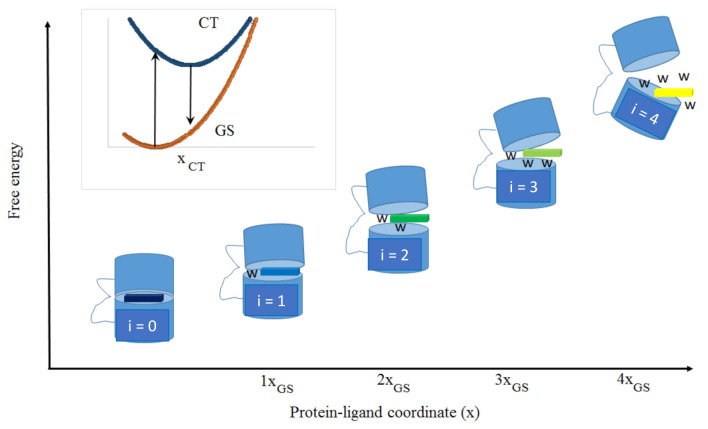
Idealized thermodynamic model for ligand–protein interactions. Free energy of protein–ligand microstates (i = 0 to 4) as a function of generalized protein–ligand coordinate (x). In this model, the protein–ligand microstates are equally spaced (*x_GS_*). The blue bi-cylindrical shapes depict the protein, ligand is represented by the colored rectangles, while bound water molecules are represented by the letter w. The change in color of the ligand along coordinate x represents the change in spectroscopic parameters of the ligand (i.e., red-shift in absorption and emission). Inset: Marcus free energy surfaces for the ligand ground state (GS) and ligand excited-state (CT) with xCT (the change in geometry of the ligand in the CT state) indicated.

**Figure 2 ijms-22-02582-f002:**
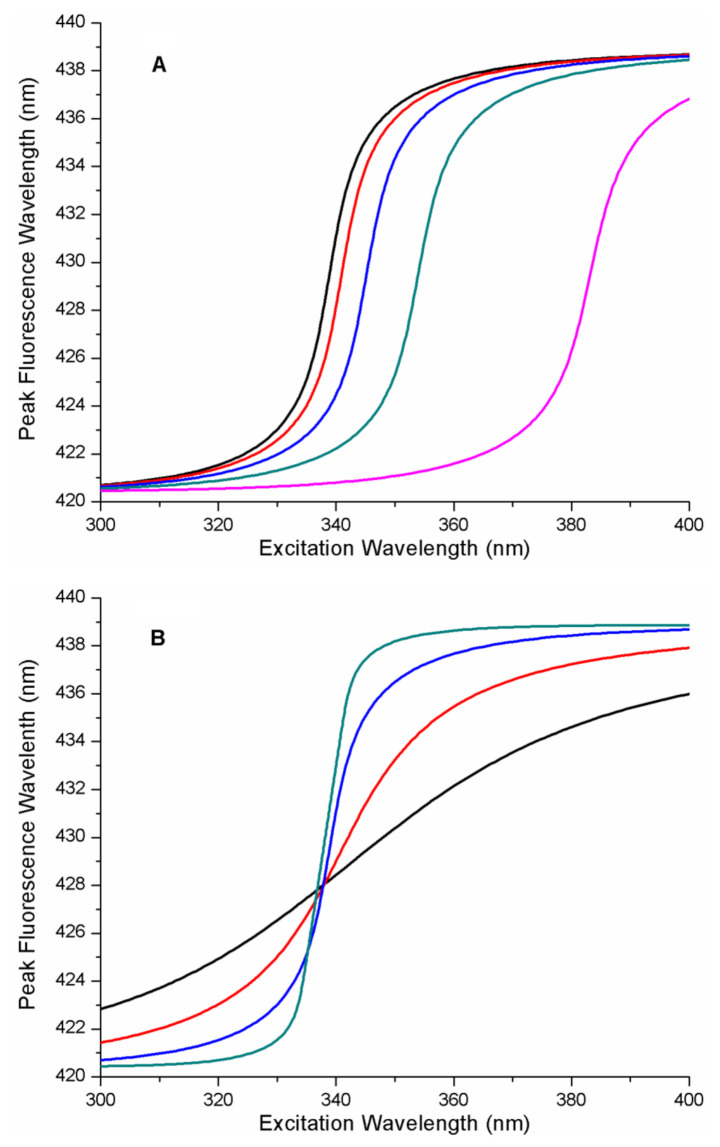
Influence of protein–ligand free energy landscape on red-edge excitation spectroscopy (REES) plots. (**A**) REES curves for a linear ramp free energy landscape (b = 1). Plots from left to right represent the effect on REES of increasing free energy gradient (A = 0.0025, A = 0.005, A = 0.01, A = 0.02, A = 0.05 eV). All other parameters were fixed (*x_GS_* = 0.1 eV^0.5^, b = 1). The photo-physical parameters for the ligand, L were fixed (*x_CT_* = 0.89 eV^0.5^, *κ* = 1, *∆G*_*CTGS*0_ = 3.346 eV, *∆G_CTGSN_* = 3.223 eV). Plots were generated with the specified parameters using Equations (13) and (14) from the theory section (the range of x was from −1 to 2). (**B**) REES curves for a linear ramp free energy landscape (b = 1). Plots represent the effect, on REES, of increasing spacing between microstate free energy surfaces along the reaction coordinate from shallowest to steepest near the transition mid-point (*x_GS_* = 0.025, *x_GS_* = 0.05, *x_GS_* = 0.1, *x_GS_* = 0.2 eV^0.5^). All other parameters were fixed (b = 1, A = 0.0025 eV). The photo-physical parameters for the ligand, L were fixed (*x_CT_* = 0.89 eV^0.5^, κ = 1, *∆G_CTGS0_* = 3.346 eV, *∆G_CTGSN_* = 3.223 eV). Plots were generated with the specified parameters using Equations (13) and (14) from the theory section (the range of x was from −1 to 2).

**Figure 3 ijms-22-02582-f003:**
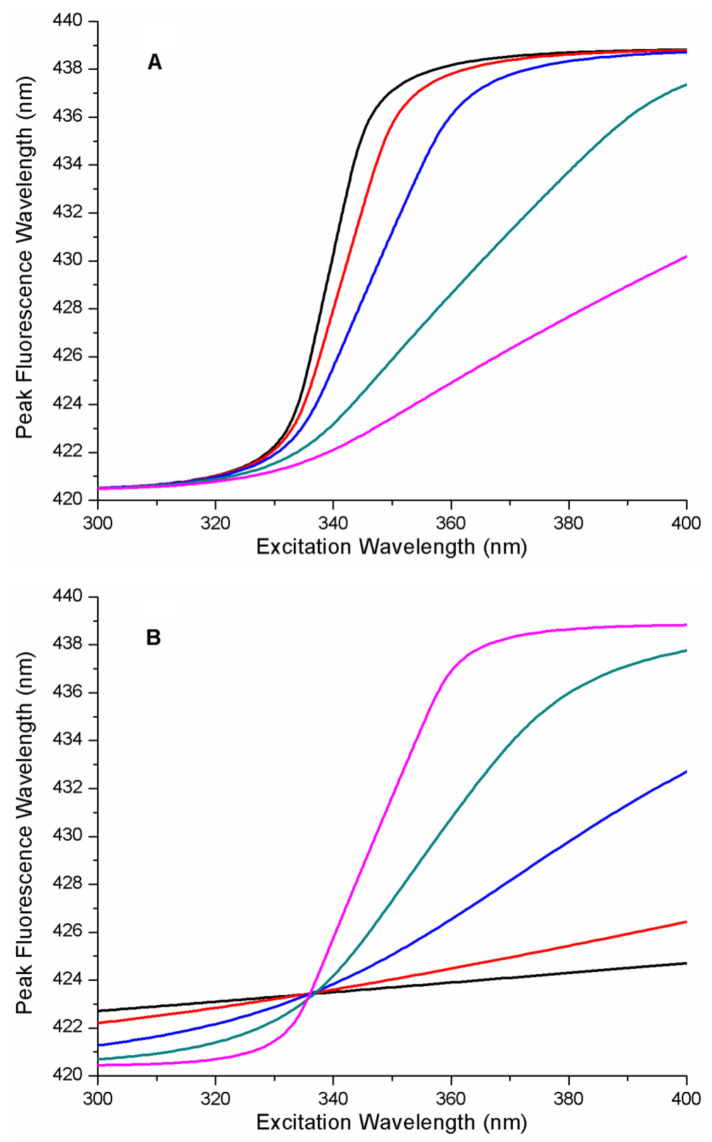
REES curves for a quadratic free energy landscape (b = 2). (**A**) Plots from left to right represent the influence of increasing the free energy parameter A, on REES curves (A = 0.001 eV, A = 0.002 eV, A = 0.005 eV, A = 0.01 eV). All other parameters were fixed (*x_GS_* = 0.15 eV^0.5^, b = 2). The photo-physical parameters for the ligand, L were fixed (*x_CT_* = 0.89 eV^0.5^, κ = 1, *∆G*_*CTGS*0_ = 3.346 eV, *∆G_CTGSN_* = 3.223 eV). Plots were generated with the specified parameters using Equations (13) and (14) from the theory section (the range of x was from −1 to 2). (**B**) Plots represent the effect on REES of increasing spacing between microstate free energy surfaces along the reaction coordinate from shallowest to steepest near the transition mid-point (*x_GS_* = 0.01, *x_GS_* = 0.02, *x_GS_* = 0.05, *x_GS_* = 0.1, *x_GS_* = 0.2 eV^0.5^). All other parameters were fixed (b = 2, A = 0.0025 eV). The photo-physical parameters for the ligand, L were fixed (*x_CT_* = 0.89 eV^0.5^, κ = 1, *∆G*_*CTGS*0_ = 3.346 eV, *∆G_CTGSN_* = 3.223 eV). Plots were generated with the specified parameters using Equations (13) and (14) from the theory section (the range of x was from −1 to 2).

**Figure 4 ijms-22-02582-f004:**
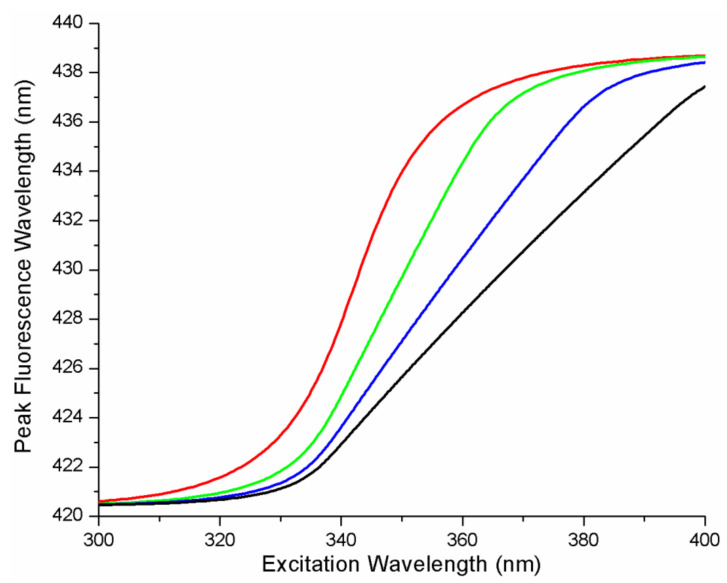
REES curves for a quadratic energy landscape (b = 2) as a function of the number of microstates, N. Plots from left to right represent the influence of increasing the number of microstates, N, on the REES curves (N = 5, N = 10, N = 15, N = 20). All other parameters were fixed (b = 2, A = 0.0025 eV, *x_GS_* = 0.15). The photo-physical parameters for the ligand, L, were fixed (*x_CT_* = 0.89, κ = 1, *∆_CTGS0_* = 3.346 eV, *∆G_CTGSN_* = 3.223 eV). Plots were generated with the specified parameters using Equations (13) and (14) from the theory section (the range of x was from −1 to 2).

**Figure 5 ijms-22-02582-f005:**
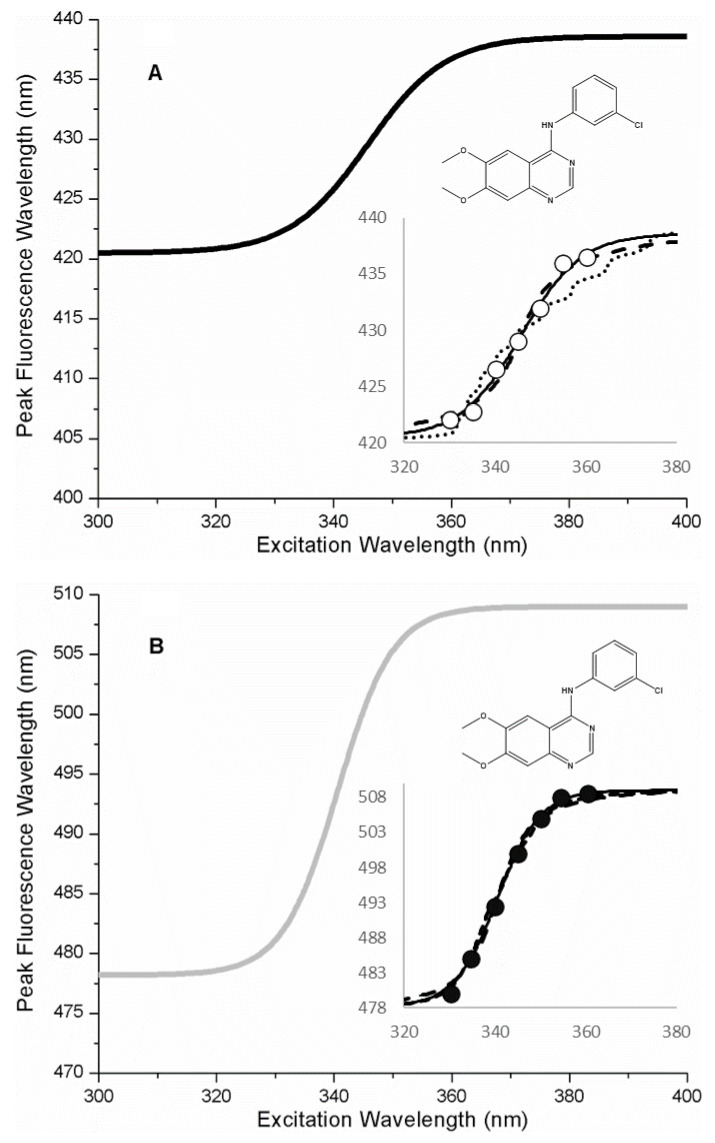
REES curves for kinase inhibitor–kinase complexes. (**A**) p38α kinase-AG1478 complex (sigmoidal fit, black symbols). Inset: REES data (open circles), fit to quadratic FEL model (solid line), fit to linear FEL model (dashed line), fit to cubic FEL model (dotted line). (**B**) APH2 aminoglycosidase-AG1478 complex (sigmoidal fit, grey line). Inset: REES data (filled circles), fit to quadratic model (solid line), fit to linear model (dashed line), fit to cubic model (dotted line). REES data (peak emission wavelength as a function of excitation wavelength) were from [17]. Structure of the ligand, AG1478, is indicated in the figure.

**Figure 6 ijms-22-02582-f006:**
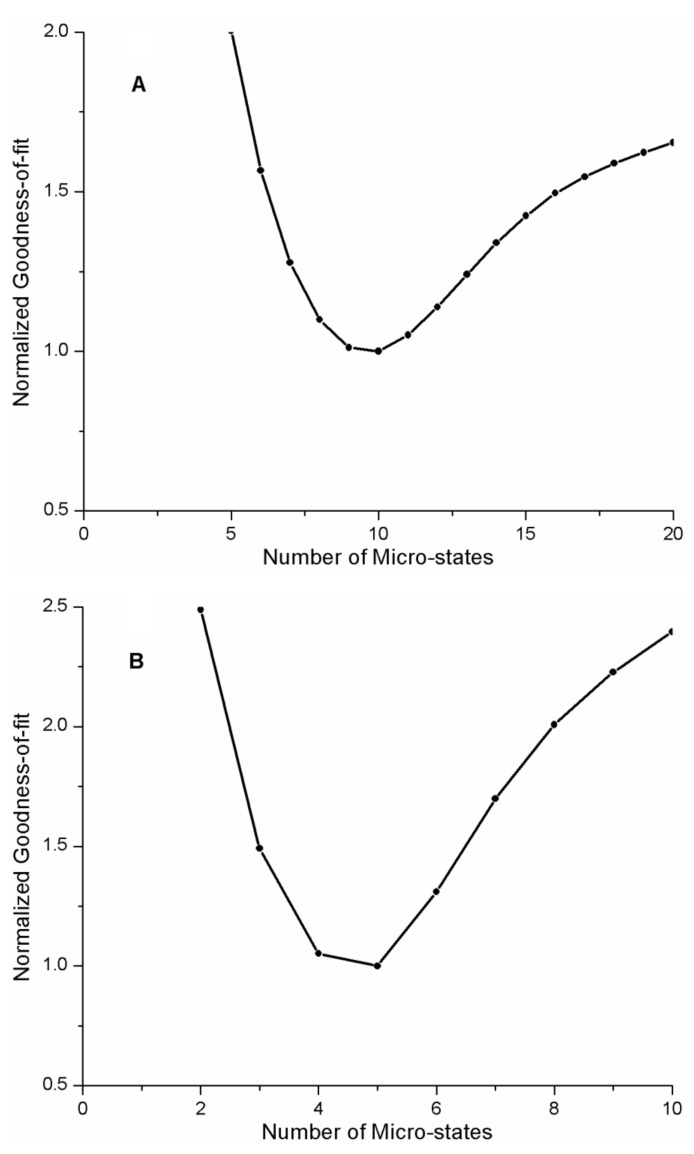
Goodness-of-fit as a function of number of microstates, N. (**A**) p38α kinase-AG1478 complex. (**B**) APH2 aminoglycosidase-AG1478 complex.

**Figure 7 ijms-22-02582-f007:**
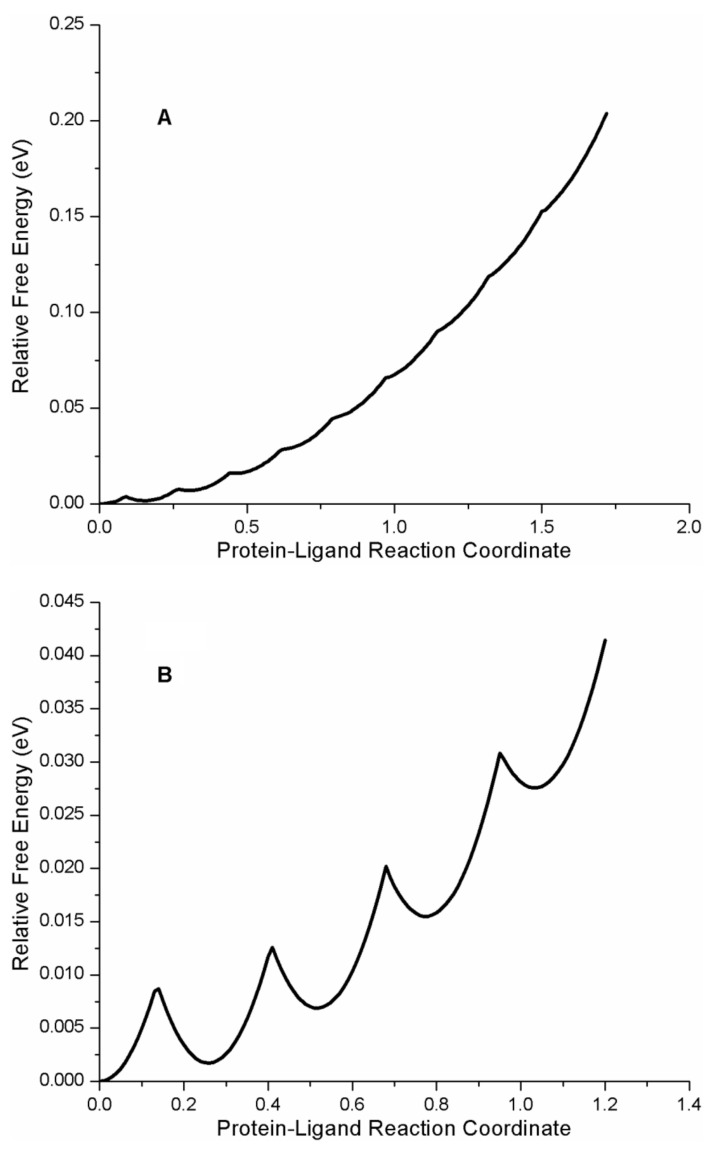
Free energy landscapes for AG1478-p38α kinase and AG1478-APH2 derived from REES data. Vertical axes represents the free energy difference (relative to zero for the most stable state) in units of eV and horizontal axis is the protein–ligand reaction coordinate, x. (**A**) FEL for AG1478-p38α kinase. (**B**) FEL for AG1478-APH2.

**Table 1 ijms-22-02582-t001:** Fit photophysical parameters for ligand from red-edge excitation spectroscopy (REES) curve. (Reorg denoted reorganization energy, other abbreviations/symbols defined in main text).

Ligand	Protein	Type FEL	*∆G*_*CTGS*0_ (eV)	*∆G_CTGSN_* (eV)	*x_CT_* (eV)^0.5^	Reorg (eV)
AG1478	P38alpha	Quadratic	3.347	3.223	0.891	0.397

**Table 2 ijms-22-02582-t002:** Fit parameters for REES curve according to different thermodynamic models. (SLSQ refers to the sum of least-squares, a measure of the quality of the model fit to the experiment).

Ligand	Protein	Type FEL	A (eV)	*x_GS_* (eV)^0.5^	*∆G_GS90_* (eV)	SLSQ
AG1478	P38alpha	Linear	0.0083	0.078	0.074	53
AG1478	P38alpha	Quadratic	0.00178	0.153	0.144	4
AG1478	P38alpha	Cubic	0.00049	0.305	0.357	205

**Table 3 ijms-22-02582-t003:** Input versus output for selected REES and free energy landscape (FEL) models.

	INPUT			OUTPUT	
*∆G_GSN0_* (eV)	N	b	*∆G_GSN0_* (eV)	N	b
0.041	10	2	0.047	8.9	2.1
0.081	10	2	0.080	9.10	1.9
0.162	10	2	0.150	9.10	1.9
0.240	10	2	0.210	9.10	2.3

**Table 4 ijms-22-02582-t004:** Fit photophysical parameters for ligand from the REES curve. (Reorg denoted reorganization energy, other parameters/symbols are defined in the main text)

Ligand	Protein	Type FEL	*∆G*_*CTGS*0_ (eV)	*∆G*_*CTGSN*_ (eV)	*x_CT_* (eV)^0.5^	Reorg (eV)
AG1478	APH	Quadratic	3.172	3.015	1.076	0.579

**Table 5 ijms-22-02582-t005:** Fit parameters for REES curve according to different thermodynamic models. (SLSQ refers to the sum of least-squares, a measure of the quality of the model fit to the experiment).

Ligand	Protein	Type FEL	A (eV)	*x_GS_* (eV)^0.5^	*∆G_GS40_* (eV)	SLSQ
AG1478	APH	Linear	0.00558	0.207	0.022	58
AG1478	APH	Quadratic	0.00172	0.258	0.028	6.01
AG1478	APH	Cubic	0.000403	0.268	0.026	55

## Data Availability

No new data were created or analyzed in this study.

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
