# Peer review of "Red-Edge Excitation Shift Spectroscopy (REES): Application to Hidden Bound States of Ligands in Protein–Ligand Complexes"

_ijms, 2021, doi:10.3390/ijms22052582_

Round 1

Reviewer 1 Report

In this manuscript, the authors developed a theoretical model of red-edge excitation spectroscopy for determining the signal arising from transient bound states of protein ligand complexes. Their model is based on Marcus theory and statistical thermodynamics and is able to simulate experimentally observed curves for kinase inhibitor-kinase complexes. This is a comprehensive and detailed manuscript that is easy to follow. I have some minor comments.

  1. Would it be possible to plot the experimental result from Ref 17 along with the simulated curves corresponding to linear, quadratic and cubic FELs?
  2. Do the authors have an explanation of why the quadradic function provided the best fit? Did they try a weighted function including quadratic and cubic?
  3. Can the authors comment more on the role of water molecules on the observed thermodynamics?
  4. How structurally different are the different microstates? Where is the largest difference?

Reviewer 2 Report

Protein fluorescence is a widely used technique in molecular biology. Since the discovery of protein fluorescence, the light absorption and fluorescence of proteins and the development of the method based on this phenomenon for investigation the structure and function of proteins themselves were developed synchronously. The wide use of the techniques based on protein fluorescence is their quite accessibility and sensitivity and to various structural changes in the protein.

The work under review also, on the one hand, represents the development of the red edge excitation shift (REES) method and, on the other hand, provides interesting information about the protein structure, which can hardly be obtained by another method. I do not like this definition of the technique, since it is not clear about what shift it is talked. In reality it is the shift of the fluorescence spectrum upon excitation at the red edge of the absorption spectrum. Although this is rather my reasoning and not a comment to the authors, since the abbreviation is already generally accepted, and of course it is better to use generally accepted terms.

The first mentioning of the phenomena of protein fluorescence spectrum shift upon excitation at the red edge of its absorption spectrum happened back in the 80s of the last century.

In those years, it was rather an interesting phenomenon, while recently its revival has been observed, since it becomes clear that this is one of the few methods that can provide information about the intermediate states of a protein. (At the same time, when this phenomenon was discovered, it was believed that proteins can have only one intermediate state.)

Now the role of the conformational lability of proteins has become evident, and this method can provide unique information. In the work under review the authors described model relating fluorescence shift at red-edge excitation with thermodynamic parameters of the protein-ligand free energy landscape and photo-physical parameters relating to the fluorescent ligand, and applied this theoretical model for the experimental data which the authors published before, namely data from the small molecule inhibitor-kinase complexes AG1478-APH and AG1478-p38.

The work is well done. I have only a few questions:

1) The approach developed by the authors is important for the study of intrinsically disordered proteins (IDPs), but the authors do not write about it, thereby cutting off a large number of potential readers who may be interested in the article. Maybe the authors should pay attention to this?

2) Everything works well when you can get good sigmoidal curves, but this is not always the case. How do the authors assess how reliable it will be in other cases?

3) If there are 10 intermediate states, may be, it is already a continuum?

Reviewer 3 Report

In general, I like the idea of the authors – it is interesting and potentially useful.

However, there are two problems with the paper that must be addressed.

The first problem is technical – when dealing with the experimental data, the authors refer to their previous work (ref 17). Hence, to understand the results one needs to read the second paper – and e.g. the data in Fig. 5 and Fig. 5d in ref 17 look different. Also the spectra in Fig. 2a in ref 17 exhibit redistribution of vibronic intensities rather than a classical REES. I strongly suggest the authors to include experimental data (at least emission spectra at different excitation and experimental data on the emission dependence of excitation) and description of the system (e.g. to show the fluorophore’s structure) to make the text more readable – it seem that this question can be solved easily.

The second problem is methodological. The authors do not explain the choice of the FEL landscape starting from Eq. (7) – why do they introduce a number of discrete states and the form Ai^b rather than some (continuous) distribution of x_gs and x_ct etc? How many parameters (exactly!) are obtained from a single REES curve? Although the model provides for logical results in modelling, I don’t think that the fitted parameters can be interpreted in terms of the number of intermediate states, as there are too many variables in the system. Even the question about the interaction between the protein and liganf is speculative (quadratic form etc – why can not one introduce some dependence of x_gs, k etc on the distance from the binding site by monotonic function?). Overall, while the equations can be moved to the SI, the authors must provide for more details on the choice of the G_gs and G_ct form in Eq. 7 and 8, maybe with some illustration, and justify the correctness of inverse problem solution.  

Round 2

Reviewer 3 Report

It seems that the authors did not understand my question or try to avoid it. I've asked "Although the model provides for logical results in modelling, I don’t think that the fitted parameters can be interpreted in terms of the number of intermediate states, as there are too many variables in the system."

Obviously, changing the number N does not change the number of parameters. Also obviously, the REES curve must be S-shaped, and fitting it to sigmoid (which, as the authors rightly claim, is determined with 4 parameters) would result in a good R^2. And definitely I didn't ask the authors why fitting of the data with a sigmoid had been performed. H

I see from the text that "The REES curve depends on the free energy landscape parameters (A and b), the microstate spacing (xGS), the number of microstates N, and L parameters". Here N is one of the parameters, and no one knows how to verify whether it is 4 or 10 -- hence, the validity of the model, which has extra parameters, can not ne proven! Hence, in terms of the positivism philosophy, such theory (hypothesis) is intrinsically unfalsifiable (https://en.wikipedia.org/wiki/Falsifiability). In other words, you have a model with dozens of assumptions (e.g. similar A and b for all states and so on), then you solve direct task (model the REES curve -- ok), and then solve the inverse task, which is incorrect (overparametrized). And from this you get e.g. N, which can not be verified. Hence, the model itself is not verified too, we don't know whether it is good or bad.

If the authors could prove that I'm wrong with my considerations, e.g. by proving that the inverse problem can be solved correctly -- I agree that the paper can be accepted for publication. But obviously the only way to to this is to introduce even more assumptions ("the spring constant is fixed k=1, the FEL type is fixed, and N is fixed (not a variable in the fit but different fits with different Ns can be compared)", as the authors say). Hence, I don't believe that the conclusions like Fig. 7 can be obtained from the REES curve because of extremely arbitrary choice of parameters and manifold of assumptions. 
